# Community-centred interventions for improving public mental health among adults from ethnic minority populations in the UK: a scoping review

Cleo Baskin ,[1] Geiske Zijlstra ,[1] Mike McGrath,[2] Caroline Lee,[3] Fiona Helen Duncan ,[4] Emily J Oliver ,[4] David Osborn,[2,5] Jen Dykxhoorn,[2,6] Eileen F S Kaner,[7] Louise LaFortune,[3] Kate R Walters ,[6] James Kirkbride ,[2] Shamini Gnani [1]

CB and GZ contributed equally.

For numbered affiliations see end of article.

**Correspondence to**
Cleo Baskin;
cleo.baskin18@imperial.ac.uk

## ABSTRACT

**Objectives** Undertake a scoping review to determine the effectiveness of community-centred interventions designed to improve the mental health and well-being of adults from ethnic minority groups in the UK.

**Methods** We searched six electronic academic databases for studies published between January 1990 and September 2019: Medline, Embase, PsychINFO, Scopus, CINAHL and Cochrane. For intervention description and data extraction we used the Preferred Reporting Items for Systematic Reviews and Meta-Analyses extension for Scoping Reviews checklist and Template for Intervention Description and Replication guide. Quality was assessed using Cochrane risk of bias tools. Grey literature results were deemed beyond the scope of this review due to the large number of interventions and lack of available outcomes data.

**Results** Of 4501 studies, 7 met the eligibility criteria of UK-based community interventions targeting mental health in adults from ethnic minority populations: four randomised controlled trials, one pre/post-pilot study, one cross-sectional study and one ethnographic study. Interventions included therapy-style sessions, peer-support groups, educational materials, gym access and a family services programme. Common components included a focus on tackling social isolation, using lay health workers from within the community, signposting and overcoming structural barriers to access. Four studies reported a statistically significant positive effect on mental health outcomes and six were appraised as having a high risk of bias. Study populations were ethnically heterogeneous and targeted people mainly from South Asia. No studies examined interventions targeting men.

**Conclusions** There is a paucity of high-quality evidence regarding community-centred interventions focused on improving public mental health among ethnic minority groups. Decision makers need scientific evidence to inform effective approaches to mitigating health disparities. Our next steps are to map promising community activities and interventions that are currently being provided to help identify emerging evidence.

## INTRODUCTION

Poor mental well-being disproportionately affects people from ethnic minority populations in the United Kingdom (UK).[1–8] Bias,

### Strengths and limitations of this study

► This scoping review addresses the large evidence gap which exists on the effectiveness of community-centred interventions in improving public mental health of UK ethnic minority populations, who are disproportionately affected by poor mental health.

► A six-stage scoping review framework, including stakeholder consultation, was applied to summarise the evidence for community-centred interventions.

► We included all community interventions irrespective of setting and type.

► Analyses of review data were limited to studies published in peer-reviewed journals and citation tracking of references.

► We used national definitions for ethnic minority categories but acknowledge challenges in viewing ethnic minority groups as homogeneous in their mental health needs.

racism and discrimination have had implications on many societal, structural and institutional risk factors for poor mental health, including worse employment,[9] housing conditions[10] and healthcare,[11] and increased likelihood to enter the criminal justice system[12] and live in poverty.[13] Stigma toward mental illness is also reported to be higher among ethnic minority groups.[14]

These risks affect mental health through numerous pathways, for example, stigma and the lack of racially, culturally and/or ethnically appropriate healthcare may cause delays in help-seeking, ultimately leading to poorer outcomes.[15 16] Additionally, stress—both acute and chronic—can affect physical health and mental well-being. The COVID-19 pandemic has exacerbated these risks, and has had a disproportionate impact on ethnic minorities.[17]

**Table 1** Eligibility criteria

| Inclusion criteria | Exclusion criteria |
| --- | --- |
| Adults aged 18–64 years | Children aged 0–17 or adults aged over 64 years |
| Individuals from Asian/Asian British, Black/ African/Caribbean/ Black British, mixed/multiple ethnic groups and Arab ethnic groups and Gypsies or Irish Travellers | Refugees, asylum seekers and all White British ethnicities other than Gypsies or Irish Travellers |
| Adults with no known mental disorder or diagnosed with a common mental disorder, such as anxiety and depression | Adults with severe mental illness and dementia |
| Community-centred interventions, that is, interventions that take place in a community setting, or in a health setting but delivered by the community and/or voluntary sector | Clinical interventions that involve treatment by a clinician in any setting |
| Study measures any domain or aspect of mental health and well-being | No outcome measures for mental health or well-being |
| All studies based on primary research | Abstracts, posters, books or chapters, editorials and letters |
| Study conducted in the UK | Study conducted outside of the UK |
| Published from 1990 onwards | Published before 1990 |
| Full-text research article available | No full-text research article available |

In the UK, the term Black, Asian and minority ethnic (BAME) is frequently used to refer to individuals who are in a racial or ethnic minority.[18] However, this umbrella term groups together a population with differential and complex risks for poor mental health.[19 20] For example, the risk of psychosis among the Black Caribbean population is nearly seven times higher than the White British population,[21] but not for South Asian groups.[22]

Strengthening public mental health involves both the promotion of mental health and well-being and the prevention of mental illness. It has the potential to effectively and sustainably reduce social and racial inequalities in mental health outcomes.[23 24] In pursuit of this, many UK initiatives have been implemented- such as the 2005 'Delivering Race Equality in Mental Healthcare Action Plan'[25] but have failed to successfully narrow the observed mental health gap.[26 27]

Community-centred interventions (ie, those that take place in a community setting, or are delivered by the community and/or voluntary sector) have potential to influence the cultural and social factors that protect and promote mental health and well-being. These include social connectedness, access to safe and affordable housing, and power in local decision-making.[28 29] In 2017, the National Health Service (NHS) England introduced local financial incentives to improve mental health outcomes for ethnic minority groups.[30] However, local decision-making has been hindered by a lack of evidence. The aim of this scoping review was to summarise the evidence for community-centred interventions focused on improving the public mental health of ethnic minority groups in the UK.

## METHODS
We undertook a scoping review, an approach for summarising evidence and identifying knowledge gaps

in unclear and emerging fields,[31 32] using Arksey and O'Malley's six-stage methodological framework.[33] The sixth stage, stakeholder consultation, was undertaken by involving stakeholders and peer researchers as part of a wider research programme.[34] Scoping reviews were not eligible for prospective registration with the international prospective register for systematic reviews at the time of conducting the review.[35]

### Study identification
A systematic search of six electronic databases between January 1990 and September 2019 was conducted: Medline, Embase, PsycINFO, Scopus, CINAHL and Cochrane. The search strategy (online supplemental appendix 1) was created with the support of a medical librarian and was based on the population, intervention, comparison, outcomes, context (PICOC) framework. It included key terms for ethnicity, age range, geography and mental health outcomes. No intervention or comparison terms were included to optimise capture of all relevant studies.

We excluded evidence from the grey literature as an initial search of primary data from local government, relevant third sector and NHS websites identified numerous activities and possible interventions (over 50 in a small geographical area). Information was provided in formats such as a flyer or a website or Facebook page describing services and activities, but with limited descriptions of the community intervention and outcomes data. Consequently, data synthesis exceeded the methodological approach of a scoping review; a mapping methodology would be more appropriate.[36]

### Eligibility criteria
Table 1 outlines the inclusion and exclusion criteria used to determine the eligibility of a study. We included studies published from 1990 onwards so that our findings would inform contemporary policy and practice. Furthermore,

systematic ethnicity data collection in the UK began with the 1991 Census.[37] We included only UK studies due to variation in how race, ethnicity and ancestry are represented in different countries. This is a consequence of each country's unique pattern of migration and its political and social context.

We included studies where participants were working-aged adults (aged 18–64 years adapted from the UK definition of 16–64 years) who were either well or had a common mental disorder, such as anxiety and depression. Common mental disorders were included to reflect the high prevalence of these conditions in the general population; one in six adults in England.[38] They often go undiagnosed and so are more amenable to interventions focused on prevention and the promotion of positive mental health.[39] Studies focusing on people with severe mental illness (including suicide and psychotic disorders), or people affected by young-onset dementia, were not included due to the need for specialist mental health treatment and tailored support. Studies that specifically targeted new migrants and refugee groups were also excluded due to the specific and complex mental health needs of this population, such as the high prevalence of post-traumatic stress disorder.[40 41] Studies were included if new migrants or refugees happened to be recruited as participants but not if the study's intervention was specifically designed for them.

Only community-centred interventions were eligible for inclusion. We defined community-centred broadly to include all interventions that were not clinical in nature, that is, did not involve the treatment of patients by health professionals such as cognitive–behavioural therapy and drug therapy. Therapy-style sessions that were delivered by lay health workers were included. There is a wide range of overlapping concepts in community-based and community-centred terms, and these were captured by our broad search strategy.

The UK Office of National Statistics 2011 Census ethnicity terms were used (table 2).[42] Ethnic groups were deemed a minority if they fell under 'Mixed', 'Asian', 'Black' and 'Other' categories. All ethnic groups under the 'White' category were excluded, except for the 'Gypsy or Irish Traveller' group due to their comparatively higher social and economic disadvantage and observed health inequalities.[8]

### Study selection

Two independent reviewers (GZ, CB) screened non-duplicate titles and abstracts against eligibility criteria (n=4501). The abstracts that matched criteria (table 1) were reviewed in full by GZ and CB (n=45). All conflicts were resolved by a third reviewer (SG); this was needed for 31% (n=14) of full-text articles. Full texts that were not freely available were accessed via the British Library. Additional articles were identified by examining references and forward–backward citation searching (figure 1). Covidence systematic review software[43] was used to remove duplicates and screen titles.

**Table 2** Ethnicity as classified by the Office for National Statistics

| | |
|---|---|
| White | English/Welsh/Scottish/Northern Irish/British |
| | Irish |
| | Gypsy or Irish Traveller |
| | Any other White background |
| Mixed/multiple ethnic groups | White and Black Caribbean |
| | White and Black African |
| | White and Asian |
| | Any other mixed/multiple ethnic background |
| Asian/Asian British | Indian |
| | Pakistani |
| | Bangladeshi |
| | Chinese |
| | Any other Asian background |
| Black/African/Caribbean/Black British | African |
| | Caribbean |
| | Any other Black/African/Caribbean background |
| Other ethnic groups | Arab |
| | Any other ethnic group |

### Charting of data

Data were extracted by two independent reviewers (GZ, CB). A data extraction framework was developed to capture key study characteristics: author, year of publication, setting, study design, population demographics, sample size, theory of change, and outcomes and results. Intervention details were extracted using the Template for Intervention Description and Replication guide: rationale, materials, procedures, provider, methods, location, timing, tailoring, modifications and assessment of intervention adherence.[44]

### Collation and summary of results

Quality appraisal was performed by two reviewers (GZ, CB) using the Cochrane tool for risk of bias for randomised controlled trials (RCTs).[45] Qualitative studies were appraised using the Cochrane Qualitative and Implementation Methods Group guidance and the Mixed-Method Appraisal Tool for mixed-methods studies.[46 47] A narrative data synthesis of the papers was conducted by identifying themes and mechanisms common to the community-centred interventions. Review findings were reported using the Preferred Reporting Items for Systematic Reviews and Meta-Analyses extension for Scoping Reviews checklist.[48]

### Patient and public involvement

The research question was informed by people's experiences and stakeholder workshops. This study did not involve the recruitment of patients, and no patients were involved in the design or conduct of the study.

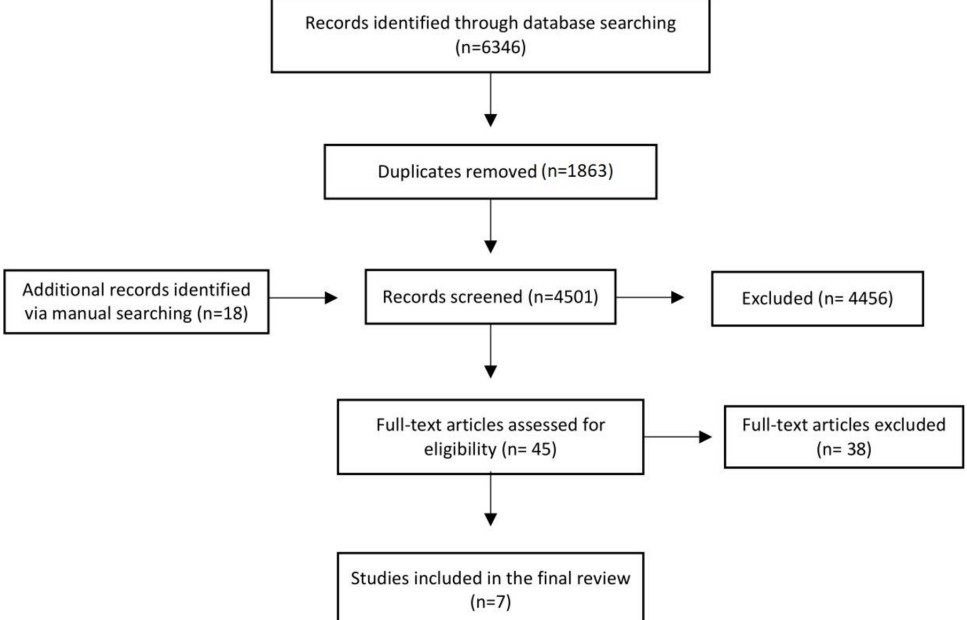

**Figure 1** Preferred Reporting Items for Systematic Reviews and Meta-Analyses (PRISMA) flow chart of search strategy. The PRISMA diagram details our search and selection process applied during the screening of articles.

## RESULTS

We identified a total of 4483 non-duplicate titles and an additional 18 from forward–backward citation searching (figure 1). Seven out of the 45 studies reviewed in full text matched the eligibility criteria and were included in this review: four RCTs (including one pilot); one pre-intervention and post-intervention pilot study; one cross-sectional and one ethnographic study (table 3).

### Intervention type and focus

The interventions were highly heterogeneous in design and focus. They included therapy-style sessions delivered by lay health workers,[49 50] peer-support groups,[51 52] an educational leaflet,[53] free gym access[54] and a family services programme.[55] All interventions were conducted in England. Certain subgroups were targeted more than others with interventions specifically tailored to individuals of Indian, Pakistani and Bengali heritage (n=3), women (n=4), and people with anxiety or depression (n=5).

Interventions were either a single session (n=1) or took place over a minimum of 10 weeks to 8 months (n=6). The duration of the family services programme is unclear. Four studies aimed to improve mental health by expanding social networks and facilitating social support[49 51 52 55] and four aimed to increase access to services by facilitating linkages or signposting to complementary social care and mental health services.[49–51 55]

Qualitative data commonly found a reduction in social isolation and stress, and improved mood and self-confidence.[50 52 54 55] One RCT reported significant improvement in social functioning[51] and two had a statistically significant positive impact on depression outcomes.[52 53] All but one study[51] were appraised as having a high risk of bias with limitations relating to small sample sizes (n=9–123, mean=43, median=30), short follow-up periods (ranging from 2 to 9 months) and participant recruitment.

### Outcome measures

Mental health outcomes were predominantly measured using standardised screening instruments for common mental disorders and social functioning (table 4). Qualitative data were predominantly collected through focus groups and semistructured interviews.[50–52 54 55] Only one study reported on costs, calculated by measuring the difference in use of health and social services between the intervention and control group.[49]

### Thematic Analysis

Following narrative synthesis, we identified three themes with the perspective of informing the delivery of public mental health interventions for ethnic minority communities: addressing social isolation and loneliness, promoting access and use of services, and being delivered by lay health workers.

### Addressing social isolation and loneliness

Chaudhry *et al*[52] evaluated the efficacy and cultural acceptability of establishing a social group for British Pakistani women with depression. Women were diagnosed with depression using the Schedule for Clinical Assessment in Neuropsychiatry and were recruited from an ongoing population-based study. Participants attended at least 6 of 10 weekly meetings at a community centre for the Pakistani community. Meetings included one session of psychoeducation and self-selected activities including

**Table 3** Summary of community-based intervention studies targeted at ethnic minority groups to improve mental health and well-being (N=7)

| Author; year | Study design | Participant characteristics | Theory of change | Intervention description | Results |
|---|---|---|---|---|---|
| Jacob et al; 2002 [53] | RCT | Asian women with depression in Ealing, London | Educate patients on depression to positively affect their perspective and outcomes of depression | Participants (n=70) either received an educational leaflet about depression or usual care | GHQ-12: recovery from common mental disorder (defined as ≤2): Odds Ratio 3.4 (95% CI: 1.01 to 11.5) at 2 months |
| Gater et al; 2010 [51] | RCT | British Pakistani women with depression in Northwest England | Improve mental health through reducing social isolation | Participants (n=123) received a social group intervention or antidepressant medication, or both | HRSD (depression): no significant effects<br>Social functioning: significant effect at 3 months for social intervention versus antidepressants, 6.1 (95% CI: 1.4 to 10.8), p=0.02, and combined group versus antidepressants, 5.9 (95% CI: 1.5 to 10.2), p=0.017 No significant effect at 9 months<br>Satisfaction: significant effects at 3 and 9 months for social versus antidepressants, 4.9 (95% CI: 2.9 to 7.0), p=0.002; 4.7 (95% CI: 2.6 to 6.7), 0.002 and combined versus antidepressants, 4.3 (95% CI: 2.6 to 6.0), p=0.001; 4.0 (95% CI: 1.8 to 6.2), p=0.005<br>Qualitative feedback found a major hindrance to participation was resistance from family members Participants engaged in the trial because of the culturally appropriate format and content of the sessions. They described their experience as 'relief from worries', 'feeling fresh' and 'better than their expectations' |
| Afuwape et al; 2010 [49] | RCT | Black adults with anxiety and/or depression in Southwark, London | Improve psychosocial functioning and feelings of hope through a culturally acceptable care package | Participants (n=40) received individual therapy and group sessions on advice on services, health education and mentoring | GHQ-29: significant effect at 3 months; adjusted mean difference 7.76 (95% CI: 0.86 to 14.65), p=0.03<br>GAF: no significant effect<br>SF-36: mental health components had significant effect, −11.93 (95% CI: −21.99 to −1.88), p=0.02<br>No effect on physical health. No significant differences in cost |
| Lovell et al; 2014 [50] | Pilot RCT | South Asian and Somalian women with moderate depression and/or anxiety in Northwest England | Improve treatment of depression and anxiety using social and community interventions | Participants (n=20) received individual or group therapy-style sessions, or usual care. Activities focused on health and well-being; signposting to service and education opportunities | CORE-OM: no effect<br>Small non-significant improvements in depression (PHQ-9), health-related quality of life (EQ-5D) and functioning (WSAS) at 5 months Qualitative data suggested that patients found the intervention acceptable, both in terms of content and delivery |

Continued

**Table 3** Continued

| Author; year | Study design | Participant characteristics | Theory of change | Intervention description | Results |
|---|---|---|---|---|---|
| Chaudhry et al; 2009[52] | Observational: pre/post- intervention | British Pakistani women diagnosed with depression In Manchester | Informal social support and mental and physical health education to reduce depression | Participants (n=9) received 10 weekly group sessions in various locations: psychoeducation, personal grooming, exercise and yoga | SRQ: reduction in depression scores pre-intervention 15 (SD=3.08) to post-intervention 11.7 (SD=5.95), p=0.039 SCAN: interviews post-intervention diagnosed 2 participants as no longer depressed Anecdotal feedback from the participants identified that the relationships developed between the participants and facilitators and the provision of transport were the most important components of the intervention |
| Rabiee et al; 2015[54] | Cross-sectional; mixed methods | Ethnic minority groups in a deprived constituency in Birmingham | Regular exercise to help improve mood, self-esteem, confidence and quality of life | Gym-for-free pilot project providing adults free access to leisure centres | Results indicate increased energy levels, confidence, mental well-being, reduction in stress and anxiety improved stress relief and anger management |
| Gray; 2003[55] | Ethnographic investigation | Ethnic minority families in Tower Hamlets, London | Reducing social isolation and poverty by improving parenting skills, self-esteem and meeting a welfare advisor | Personalised family support workers; trained volunteers of similar ethnic and cultural backgrounds | Family case records and interviews suggest close relationships were formed with the support workers of the same ethnic and cultural identity. They also portray a reduction in social isolation, bullying and racism increased advocacy with other professionals and access to relevant services |

CORE-OM, Clinical Outcomes in Routine Evaluation-Outcome Measure Scale; GAF, Global Assessment of Functioning; GHQ-12, General Health Questionnaire-12; GHQ-29, General Health Questionnaire-29; HRSD, Hamilton Rating Scale for Depression; PHQ-9, Patient Health Questionnaire; RCT, randomised controlled trial; SCAN, Schedule for Clinical Assessment in Neuropsychiatry; SF-36, Short Form survey 36; SRQ, Self-Reporting Questionnaire; WSAS, Work and Social Adjustment Scale.

personal grooming, exercise, yoga, and visits to museums and local shopping malls.

The intervention was successful in improving mental health outcomes; a significant reduction in depressive symptoms was found at the end of the 10 sessions (pre-intervention: 15, SD=3.08; post-intervention: 11.7, SD=5.95, p=0.039) and three women reported a reduction in suicidal ideas. These outcomes, and participants' anecdotal feedback, suggest that a reduction in depressive symptoms is partly attributed to reducing social isolation by bringing the women together. Participants highly valued the provision of free transport by female Urdu speakers to avoid any objections from family and community members. Heterogeneity in the content and attendance of each session makes it difficult to assess the mechanism for improved outcomes. The small sample size and lack of control group further limit the generalisability of these findings.

Chaudhry's pilot study formed the basis of the RCT conducted by Gater et al.[51] British Pakistani women (n=123) with depression were recruited from general practice surgeries. The women were randomised into one of three intervention arms: a social group intervention, antidepressant medication, or both the social group intervention and antidepressant medication. The social group involved 10 women, who attended 10 weekly sessions of social activities at a community centre. Facilitators were Urdu-speaking women who had completed a 5-day training programme.

The intervention had no significant change in depression in all three arms. Social functioning significantly improved in both the social intervention and combined treatment groups at 3 months (6.1, 95% CI: 1.4 to 10.8 and 5.9, 95% CI: 1.5 to 10.2) but not at 9 months. Focus groups, held at the end of each session, found that the intervention was too short and that a major barrier to participating was resistance from family members. Participants found it helpful to confide in others and were intending to maintain new friendships. Overall, this study was appraised as low risk and provides evidence in favour of social group interventions for short-term improvements of positive mental health.

### Promoting access and use of services

Gray[55] evaluates a Family Support Service which provides home-based support and parenting lessons for families; support for parents with a severe mental illness; and help from support workers and the local authority in child protection. The service was delivered by support workers

**Table 4** Mental health and well-being outcome measures reported in the included studies (n=7)

| Author; year | Quantitative measures | Qualitative outcomes |
|---|---|---|
| Jacob et al; 2002[53] | ▶ GHQ-12: General Health Questionnaire-12<br>▶ Short Explanatory Model Interview: self-reported perspective on depression | None |
| Gater et al; 2010[51] | ▶ Hamilton Rating Scale for Depression<br>▶ Verona Service Satisfaction Scale (adapted)<br>▶ Social functioning (measured using a tool specifically created for British Pakistani women) | Feedback forms and focus groups |
| Afuwape et al; 2010[49] | ▶ GHQ-28<br>▶ Global Assessment of Functioning<br>▶ Short Form survey 36 (measuring quality of life)<br>▶ Life Events and Difficulties Schedule | None |
| Lovell et al; 2014[50] | ▶ Patient Health Questionnaire-9 (measuring depression)<br>▶ Clinical Outcomes in Routine Evaluation-Outcome Measure Scale (measuring global distress)<br>▶ Generalised Anxiety Disorder Assessment-7<br>▶ Work and Social Adjustment Scale<br>▶ EQ-5D (measuring health-related quality of life) | Semistructured interviews regarding the acceptability of the intervention |
| Chaudhry et al; 2009[52] | ▶ Self-Reporting Questionnaire (Urdu version; screening for mental disorders)<br>▶ Schedule for Clinical Assessment in Neuropsychiatry | Anecdotal feedback collected by facilitators |
| Rabiee et al; 2015[54] | ▶ Self-completed questionnaire evaluating experience in accessing services and perceived changes in health and well-being | Focus groups analysed using established frameworks and guidelines[76 77] |
| Gray; 2003[55] | None | Family case records and interviews analysed using established frameworks[78 79] |

who were recruited from the local community or had similar backgrounds to the client group.

Thirty case records of families were examined, and 22 interviews were conducted with support workers, families themselves, and professionals managing or referring to these services. Support workers were found to have had a positive impact on families by bringing them together; reducing social isolation, bullying and racism; promoting equality of access and pertinent use of services; and liaising successfully with other health and social care services. It was highly valued that projects were culturally sensitive and that support workers were of the same ethnicity. Power dynamics were important as facilitators' decisions were of potentially significant consequence to the families. This makes the support workers not only essential to delivering the intervention but also integral to achieving a positive outcome for families.

Jacob et al[53] evaluated the effect of a culturally appropriate educational leaflet on depression delivered in general practice to women of Indian, Bangladeshi and Pakistani heritage (n=70). All participants had a common mental disorder, diagnosed through the General Health Questionnaire (GHQ). Participants had their perspective (explanatory models) on depression assessed using the Short Explanatory Model Interview (SEMI). They were then randomly assigned to receive either a leaflet about the nature, causes, prevalence and treatment of depression or to usual care from their general practitioner. The leaflet was available in English, Hindi and Punjabi and read out to illiterate participants. After 2 months,

all participants repeated the GHQ and rediscussed the SEMI, particularly focusing on the idea of depression as an illness and whether medical help is necessary.

There was no statistically significant difference between the intervention and control group on the explanatory model of depression at follow-up. The number of women who recovered from depression (defined as GHQ-12 ≤2) at 2 months was significantly associated with receiving the educational leaflet (42.9% vs 20% for controls, p<0.05; no median score of CIs was reported). However, women in the intervention group had significantly lower baseline levels of psychiatric morbidity at entry into the trial, potentially explaining the observed differences between the two groups.

Rabiee et al[54] conducted a survey questionnaire (n=257) and focus groups to evaluate the physical and mental health impact of a 'gym for free' scheme in four leisure centres. The survey was intended to elicit participants' experiences of accessing the service as well as information about their height and weight and perceived changes in their general health and well-being. Benefits of the scheme included improved confidence (n=92) and energy (n=121), reduced stress (n=77) and improved anger management (n=8).

Only nine participants participated in three focus groups: two men (one White and one Black) and seven women (four Pakistani, two Indian and one White). Participants valued building their social networks through the scheme and acknowledged the links between their physical and mental well-being. Although the authors suggest

a financial barrier inhibiting people joining the gym, no formal data were collected on cost or participant finances. The generalisability of these findings is limited as participants were selected through opportunistic sampling and may over-represent younger adults who are motivated to use the gym.

### Delivered by lay health workers

Afuwape *et al*[49] evaluated the London Cares of Life Project which is a complex social intervention designed to improve the mental health of adults of Black African or Black Caribbean origin (n=40). Participants with anxiety and/or depression, diagnosed using the WHO checklist criteria, were randomised to receive the intervention immediately or to a 3-month waiting list control and information on local mental health services. During the 3-month intervention, community health workers (psychology graduates trained to deliver the intervention) provided practical advice to address social needs and health education, and gave brief therapies based on cognitive–behavioural therapy principles.

Those who received the intervention showed significant improvement at 3 months in depressive symptoms (adjusted mean difference=7.76, 95% CI: 0.86 to 14.65) but not in general functioning (adjusted mean difference=−0.78, 95% CI: −10.40 to 8.84). The effects beyond 3 months are unknown. There was no difference in costings between the intervention and control group; health and social care utilisation costs were calculated for 3 months. The risk of bias is increased by the small sample size and participants in the non-intervention arm having a significantly greater prevalence of psychiatric history. The generalisability to the wider ethnic minority population is potentially limited due to the exclusion of non-English-speaking participants and the use of screening tools that do not have clear evidence of validity in these populations.

Lovell *et al*[50] examined the effect of a well-being intervention, based on cognitive–behavioural strategies, aimed at decreasing anxiety, depression and social isolation in both ethnic minority and elderly populations. Only results from the ethnic minority group were included in this analysis (aged 21–58 years, n=20). Participants chose between receiving the intervention individually, as part of a group, or as signposting sessions. Interventions in all three arms were delivered by well-being facilitators, after a 3-day training programme, who liaised with the participants' health and social care professionals. Individual interventions comprised eight 30-minute sessions over 16 weeks, groups were delivered 8–10 weekly 2-hour sessions, and those who participated in signposting sessions were seen three times over a 16-week period.

This intervention had no statistically significant effect on mental health outcomes. The semistructured interviews established that acceptability was enhanced by involving family and the community, having a health professional's recommendation, and having an empathetic facilitator. This study is limited by its small sample size and by having patients choose their intervention arm, albeit reflecting the realities of engagement and participation when delivering interventions.

## DISCUSSION
### Main findings

We found seven studies of community-centred interventions that aimed to improve mental health and well-being among ethnic minority groups in the UK. Although studies varied in design, four key interventional characteristics emerged.

First, interventions aimed to address social isolation through building peer-to-peer support and social networks.[49 51 52 54] Evidence suggests ethnic minority populations find it easier to start conversations on mental health within their own cultural networks rather than with health professionals; formal mental health services perceived as a last resort.[56 57] We found that social support interventions mainly targeted women and it is unclear whether men would find these acceptable.

Second, interventions aimed to overcome structural barriers in accessing care. Addressing practical considerations, such as translating educational materials into different languages[53] and providing appropriate transport,[51 52] increased participation in the intervention/activities.

Third, interventions were delivered by lay health workers.[49 50 55] Over recent years, task-shifting in healthcare from health professionals to lay health workers has been common due to the need to meet escalating demand for care.[60] Lay health workers from the same community as patients may be perceived as more accessible[56] and help reduce the associated stigma of accessing mental health services.[61 62] They may also overcome language and cultural barriers that ethnic minority groups face when trying to communicate mental health needs to their healthcare provider.[63 64] However, lay health workers, unless they have received anti-stigma training, may hold beliefs that are more stigmatising than those held by health professionals.[65]

Fourth, interventions had an emphasis on signposting and or facilitating linkages to complementary or additional services.[49–51 55] Signposting has been widely adopted in England through social prescribing schemes, but to date there has been little emphasis on cultural appropriateness.[66] Evidence suggests that mental health services provided by the voluntary and community sector and embedded in communities increase trust among ethnic minority communities, which in turn promote awareness of mental health problems and access to mental health services.[2]

Of the community interventions we identified, we found more targeted South Asian ethnic groups, which may reflect these groups making up a greater proportion of the UK population. There were no interventions designed for and targeting men, despite men being less likely to seek help for common mental health problems.[67]

## Limitations

A limitation of this study was the omission of grey literature. Preliminary grey literature searching found over 50 community-centred services, which confirms that most relevant interventions reside outside publication in peer-reviewed journals and indicates significant publication bias. We were unable to include these interventions in this review due to a large number of individual activities/interventions and the lack of available primary data. We consider a mapping methodology to be more appropriate to comprehensively summarise this evidence.[36] We also only included community-centred interventions that targeted ethnic minority communities. It is possible that ethnic minority groups access public mental health interventions through universal provision, however uptake of screening programmes and utilisation of mental health-care services show ethnic minority populations have lower use.[68]

A further limitation is the generalisability of our results that include all ethnic minority populations in the UK; homogeneous recommendations cannot be made to a culturally and ethnically heterogeneous population. Furthermore, we excluded all white ethnic minority groups apart from Irish Travellers and Gypsies. However, the UK has a white ethnic minority European population, such as people from Poland, who may be subject to similar systemic health inequalities and challenges to their mental health and well-being as other ethnic minority groups.[2]

Interventions specifically targeting refugees, asylum seekers and new migrants were also excluded. In the UK, refugees make up approximately 0.2%[69] of the population and new migrants 1.1%,[70] and research indicates that these groups are more likely to require specialised clinical interventions focused on reducing psychological trauma.[71 72] However, due to the overall paucity of evidence in this field and the likelihood of shared need, future research should consider whether this exclusion is necessary.

## Implications for policymakers and future research

We recognise that the evidence for ethnic differences in some common mental disorders is complex, variable and indeed equivocal for specific ethnic groups. However, there is a disconnect between the scale of community-centred intervention provisions and published evidence. This may imply a bias in terms of scientific priorities and funding that needs urgent rectification, as current evidence is limited and weak.

It is important that future research seeks to understand *how* successful interventions work to improve mental health and for whom, taking into account intersectionality such as between gender and ethnicity. This requires resources to be appropriately allocated to third sector organisations, which typically provide these interventions, as they are often bound to deliver short-term projects and lack resources for robust evaluations. Evaluations should also include economic analysis; community interventions are potentially an affordable means to improving public mental health.[73]

These findings also suggest the need for more activity on preventing mental illness and promoting well-being; most of the studies focused on people with existing common mental conditions. Health promotion and primary prevention alongside universal approaches are critical components of strong public mental health and sustainable health systems.[68] Additionally, further research is needed to understand the societal, structural and institutional challenges affecting community-centred public mental health interventions for ethnic minority groups to help identify potential solutions.

Lastly, labels such as 'Asian' or 'Black African' present challenges to mental health research by viewing ethnic minority groups as homogeneous in their mental health needs, despite evidence indicating otherwise.[2 74 75] We recommend that national census categories are consistently used across all sectors, so that important variations and inequities can be identified and investigated further.

## CONCLUSION

Despite well-documented ethnic disparities in mental health, there is a paucity of high-quality evidence regarding community-centred interventions that focus on improving public mental health among ethnic minority groups. Decision makers need scientific evidence to help commission appropriate services and to inform effective approaches to mitigating these health disparities. Our next steps are to map the promising community activities and interventions that are currently being provided to help identify emerging evidence.

**Author affiliations**
[1]Department of Primary Care and Public Health, School of Public Health, Imperial College London, London, UK
[2]Division of Psychiatry, UCL, London, UK
[3]Cambridge Institute of Public Health, University of Cambridge, Cambridge, UK
[4]Department of Sport and Exercise Sciences, Durham University, Durham, UK
[5]Camden and Islington NHS Foundation Trust, London, UK
[6]Department of Primary Care and Population Health, UCL, London, UK
[7]Population Health Sciences Institute, Newcastle University, Newcastle upon Tyne, UK

**Acknowledgements** The authors thank the library staff at Imperial College London for their insights in the design of search terms and their support in the use of resources.

**Contributors** SG conceived the idea. GZ, CB and SG designed and conducted the study, which was supervised by SG. GZ and CB wrote the original draft of the manuscript. MM, CL, FHD, EJO, DO, JD, EFSK, LL, KRW and JK contributed to the interpretation of the results and revised the manuscript critically for intellectual content and technical and language edits. Further reviewing and editing were undertaken by GZ, CB and SG.

**Funding** This study is funded by the National Institute for Health Research (NIHR) School for Public Health Research (SPHR) (grant reference number PD-SPH-2015). The NIHR School for Public Health Research is a partnership between the Universities of Sheffield, Bristol, Cambridge, Imperial and University College London; the London School for Hygiene and Tropical Medicine (LSHTM); LiLaC—a collaboration between the Universities of Liverpool and Lancaster; and Fuse—the

Centre for Translational Research in Public Health, a collaboration between Newcastle, Durham, Northumbria, Sunderland and Teesside Universities.

**Disclaimer** The views expressed are those of the author(s) and not necessarily those of the NIHR or the Department of Health and Social Care.

**Competing interests** None declared.

**Patient consent for publication** Not required.

**Provenance and peer review** Not commissioned; externally peer reviewed.

**Data availability statement** Data sharing not applicable as no datasets generated and/or analysed for this study. This was a scoping review with no original individual participant data.

**ORCID iDs**
Cleo Baskin http://orcid.org/0000-0001-6254-8707
Geiske Zijlstra http://orcid.org/0000-0003-2173-2430
Fiona Helen Duncan http://orcid.org/0000-0002-4929-5685
Emily J Oliver http://orcid.org/0000-0002-1795-8448
Kate R Walters http://orcid.org/0000-0003-2173-2430
James Kirkbride http://orcid.org/0000-0003-3401-0824
Shamini Gnani http://orcid.org/0000-0001-6246-9590

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
