## [Reviewer comments · BMJ Open]

ARTICLE DETAILS

TITLE (PROVISIONAL)	Community-centred interventions for improving public mental health among adults from minority ethnic populations in the United Kingdom: a scoping review
AUTHORS	Baskin, Cleo; Zijlstra, Geiske; McGrath, Mike; Lee, C; Duncan, Fiona; Oliver, Emily; Osborn, David; Dykxhoorn, Jen; Kaner, Eileen; LaFortune, Louise; Walters, Kate; Kirkbride, James; Gnani, Shamini

VERSION 1 – REVIEW

REVIEWER	Nhi-Ha Trinh Harvard Medical School, US
REVIEW RETURNED	22-Jul-2020

GENERAL COMMENTS	This is a very important topic using a new method of review called the scoping review. 1) The title is a bit deceiving: "Improving public mental health"--when actually the review looks at depressive and anxiety symptoms primarily--this title should be revised accordingly, as the exclusion criteria eliminated programs focusing on suicide and psychotic disorders, which are arguably also a part of public mental health. 2) In addition, although the review focuses on "BAME" populations and there is some glancing mention of the "cultural appropriateness" of the interventions, there is no framing of the cultural and racial context for BAME populations in the UK, or the challenges (racially, ethnically, etc) these populations face based on their minority status. 3) Relatedly, the importance of these interventions would be their focus on cultural and social factors for BAME populations, and this should be highlighted more clearly throughout the manuscript. 4) The scope of the review excludes programs from "grey" literature, but the small number of included studies (only 7) In contrast to more than 50 programs that were excluded indicates there is significant publication bias in play. More discussion about how these excluded programs differ (in addition to exclusion based on lack of evaluation) is in order.
---

REVIEWER	Katarzyna Karolina Machaczek Sheffield Hallam University, Advanced Wellbeing Research Centre (AWRC)
REVIEW RETURNED	03-Aug-2020

GENERAL COMMENTS	The manuscript details a scoping review of community-centred interventions to improve the mental health and wellbeing of adults from Black, Asian and Minority Ethnic (BAME) populations in the UK. It systematically maps literature available on the topic, it's clearly conceived and well-written. The manuscript is of great interest and adds to the current understanding of intervention design, implementation and evaluation. I have some relatively minor comments and suggestions for the authors. Abstract Lines 56 &57 'some BAME groups' I suggest the authors provide more details here. Some BAME groups such as... Strengths and limitations of the study Line 31 "This scoping review addresses the large evidence gap which exists on the effectiveness of community-centred interventions in improving public mental health of UK black and minority ethnic (BAME) populations, whom are disproportionately affected by poor mental health' For 'whom' rather than 'who' to be right in this sentence the authors would need to say "poor mental health affected whom" or something like that. I suggest the authors change 'whom' to 'who' – it's always more idiomatic. Line 47 "Peer review" - please add hyphen https://dictionary.cambridge.org/dictionary/english/peer-reviewed Introduction Line 16-26 "Further, the long-standing societal disadvantage and discrimination towards these groups has demonstrable consequences on mental health outcomes, for example through overrepresentation in the criminal justice system (13); lower uptake and poorer satisfaction with care(14); and poorer recovery rates and less likelihood to be in employment(15,16)." This sentence could be improved by dividing the mechanisms that may lead to poor mental health outcomes (e.g. lower uptake of treatment) and outcomes themselves (e.g. poorer recovery rates). Methods Line 11 ...by involving stakeholder and peer researchers as part of a wider research programme (36). – please change 'stakeholder' to 'stakeholders' Line 15 "The public were not involved in the design or conduct our research. "please add 'of: The public were not involved in the design or conduct of our research I'm curious whether the authors used appropriate frameworks (e.g. PICOS) to the literature search process? Table 1 suggests they did. Could the authors add a sentence or two to explain this – to aid clarity for the reader? Line 28 Literature searches: Please provide justification for a start date of 1990 Table 1 Eligibility criteria
--

	Page 9 Line 42 “All studies based on primary research,” please remove comma Results – addressing social isolation and loneliness Page 17 Line 45 please change “participant’s’ to ‘participants’ Page 18 Lines 28-33 please correct punctuation in this sentence: “There was significantly greater improvement in social functioning in the social intervention and combined treatment groups at three months (6.1, 95% CI: 1.4,10.8 and 5.9, 95% CI:1.5,10.2), however this effect did not persist at 9 months.” Page 20 line 47 “Jacob et al. (56) conducted an RCT in one general practice in Ealing, London evaluating the effect of a culturally-appropriate educational leaflet on depression for women of Indian, Bangladeshi and Pakistani heritage (n=70).” there is a coma missing in this sentence Results – lay health workers delivering complex interventions Page 21 line 17-18 “Although the authors suggest a financial barrier es that inhibits people joining the gym, no formal data were collected on cost or participant finances.” ‘es’ seems to be redundant Page 21 line 42 Please add ‘the’ before “World Health Organisation “ Discussion – main findings Page 26 line 22 please add “s” to increase ... mental health services increase The authors may want to consider using realist approaches in their future studies on this topic. This would allow them to utilise evidence they didn’t include in the present manuscript.
--	---

REVIEWER	Masahito Fushimi Akita University Health Center, Akita University, Japan
REVIEW RETURNED	27-Sep-2020

GENERAL COMMENTS	This study is based on the results extracted from a vast amount of literature concerning minorities. It makes several useful contributions to the study of minorities, but there are a few issues which require further attention, however. Regarding the issues related to the generalizability of the study results, since BAME covers a broad and diverse range of groups, it is difficult to conduct research in this field under a unified design. Therefore, as the authors correctly state, it is difficult to find high-quality evidence. I think this problem also applies to the results of this study. In other words, I think it is difficult to use the results of this study as evidence applicable to the many diverse minorities worldwide. In this study, a significant amount of literature was extracted in order to provide recommendations to those conducting intervention studies. Meanwhile, many studies were not included in this present study because of problems such as the scale of the study and the observation period. I think that it will be a valuable contribution to intervention studies in the future if you describe and advise on the size, duration, and type of study designs by which intervention studies may be conducted. I further suggest you present these recommendations to researchers from the standpoint of scoping reviews. I suggest you clarify this point in your paper.
--

	The third part of your Discussion states that lay health workers' activities can help reduce stigma. While I agree with this position, the possibility that non-professionals (including lay health workers) may have more stigma than professionals cannot be ruled out. Based on this viewpoint, it is also necessary to cite any literature that identifies the danger of intervention by non-professionals. This study excludes new migrants and refugee groups. Although this decision was informed by the high prevalence of PTSD and other conditions in these groups, it is widely regarded that a high incidence of traumatic experiences is the general case among BAME groups. Moreover, new migrants and refugees also require not only professional care but care (from lay health workers, etc.); hence, it is appropriate to consider whether their exclusion was necessary or not. Other: On a minor note, you use the term "black and minority ethnic (BAME)" in the Strengths and Limitations section of your study, while in other parts you use the term "Black, Asian, and Minority Ethnic (BAME)". This should be corrected for uniformity and accuracy.
--	---

REVIEWER	Haider Mannan Western Sydney University Australia
REVIEW RETURNED	22-Oct-2020

GENERAL COMMENTS	This article on scoping review is in an area where there is a gap in the relevant area for UK. The article is stated this gap. The aims are clear also. The structure of the article is ok. The pros and cons of the study are also well addressed. However, I feel the writing could be improved. Therefore, I suggest a resubmission with a minor revision. The article requires proof reading before resubmission.
---

REVIEWER	Genevive Meredith Cornell University, USA
REVIEW RETURNED	04-Dec-2020

GENERAL COMMENTS	 - Consider adding "medical" before care: page 5, line 23 - Based on statement (page 7, line 47), consider adding assessment of grey literature to next steps - could be valuable info re: intervention selection, processes used, and facilitators or barriers - Page 8, line 14. May be worth noting why translation of studies from other countries akin to UK was not considered - Page 8, line 30-37: I found this sentence hard to understand - may be missing a word, or re-work punctuation? - Before page 10, line 37 (II Study selection) - should you list how many articles were identified via the web search, even though you also list in results? I ask because you note a % in the next pp. - Page 10, line 39 - say more here? screened titles and abstracts for what? what made an article eligible for full review? When did SG come in (title/abstract or full review?) - Page 10, line 50- presumably duplicates were removed before reading all? - Page 12, line 11 - you note getting from 45 studies to 7. How did this happen? Not clear from methods section.
---

	 - Page 12, line 18 - add a paragraph about the demographics of people in the 7 studies? Later you note more women than men - may be worth stating up front. - Page 12, line 20 - heterogeneous in what respect? Also, maybe add a sub-heading of 'intervention type and focus' here? - Page 12, line 36-46 - given the small number of studies, consider phrasing this in another way. one had.... three had... the majority had... the median or mean was... - Page 14 - put the article's reference number in column 1 for ease of reference? - Page 16 - you show three categories of outcome measures in this section (addressing.... Promoting... Lay...) - how did you decide on these? Emergent coding? maybe describe in methods? Is this narrative a synthesis of the papers, or an independent (your) assessment of the data and the bias? May want to make that clear. (It almost reads like a narrative of the table - not sure that it adds much value?)  might you re-frame this section per your research question: did this intervention improve the mental health and wellbeing of population: y/n, why/why not, lessons to take away - Page 24 - somehow I seem to want to see reference to your study papers also shown here in support of the claims/findings you note. Which studies showed interventions aimed at addressing social isolation? etc. Also, I think this section describes the characteristics of interventions... Was that the focus of your research? Or, were you looking at what types of interventions actually improve MH? If the latter, perhaps organize the discussion to compare and contrast the characteristics between interventions that work vs. don't?
--	---

VERSION 1 – AUTHOR RESPONSE

BMJ Open Reviewer's comments	Responses
Reviewer 1: Nhi-Ha Trinh, Harvard Medical School, US	
Title: The title is a bit deceiving: "Improving public mental health"--when actually the review looks at depressive and anxiety symptoms primarily--this title should be revised accordingly, as the exclusion criteria eliminated programs focusing on suicide and psychotic disorders, which are arguably also a part of public mental health. [Page 4, lines 25-26 & 45-46; page 7; lines 35-41; page 26; lines 93-96]	Thank you for your comment. We respectfully disagree with amending our title as it reflects the intended scope of our review. We agree public mental health encompasses more than depressive and anxiety symptoms. However, our review has highlighted that current research in the field of public mental health is focussed on mental illness. Therefore, to clarify the scope of our review, we have made the following revisions: Introduction (page 4; lines 25-26) we have added the sentence:

	"Strengthening public mental health involves both the promotion of mental health and wellbeing and the prevention of mental illness." Discussion (page 26; lines 93-96) we have addressed the research focus on common mental conditions: "Most of the studies focussed on people with existing common mental conditions. Health promotion and primary prevention alongside universal approaches are critical components of strong public mental health and sustainable health systems (74)." Methods page 7; lines 35-41 we have addressed excluding severe mental illness: "Common mental disorders were included to reflect the high prevalence of these conditions, which is one in six adults in England (37). They often go undiagnosed and so are more amenable to interventions focused on prevention and the promotion of positive mental health (38). Studies focusing on people with severe mental illness (including suicide and psychotic disorders), or people affected by young onset dementia, were not included due to the need for specialist mental health treatment and tailored support."
General comment: Although the review focuses on "BAME" populations and there is some glancing mention of the "cultural appropriateness" of the interventions, there is no framing of the cultural and racial context for BAME populations in the UK, or the challenges (racially, ethnically, etc) these populations face based on their minority status. [Page 4, lines 3-16]	Thank you for your comment. We have revised the first paragraph (page 4; lines 3-16) to add depth and context to the cultural and racial challenges experienced by BAME populations in the UK.
General comment: The importance of these interventions would be their focus on cultural and social factors for BAME populations, and this should be highlighted more clearly throughout the manuscript.	Thank you for your comment. The studies we identified did not focus in investigating the underlying mechanisms of interventions with respect to their influence on cultural and societal factors. However, we agree that the importance of community-based public mental health interventions on these factors should be better

highlighted in the manuscript. Therefore, we have made the following revisions:

Introduction (page 5; lines 32-35) we have revised the sentence to read:

“Community-centred interventions (i.e. those that take place in a community setting or are delivered by the community and/or voluntary sector) are increasingly recognised to have potential to influence the **cultural and social factors** that protect and promote mental health and wellbeing.”

Discussion (page 23; lines 14-19) we have revised the paragraph to read:

“Second, interventions aimed to overcome cultural and other barriers in accessing care. For example, by translating educational materials (55), addressing practical considerations, such as travel(53,54) and using facilitators that shared the same ethnicity and/or life experience as participants (53-55,57). This concurs with emerging evidence that culturally adapted interventions lead to better outcomes among minority ethnic groups (64,65).

Implications for Policy (page 26; lines 84-86) we have revised the paragraph to read:

“It is important that future research seeks to understand *how* successful interventions work to improve mental health and for whom, taking into account intersectionality such as between gender and ethnicity.”

Implications for Policy (page 26; lines 96-99) we have added the following paragraph:

“Additionally, further research is needed to understand the societal, structural and institutional challenges affecting community-centred public mental health interventions for BAME groups to help identify potential solutions.

General comment: The scope of the review excludes programs from "grey" literature, but the small number of included studies (only 7) In contrast to more than 50 programs that were excluded indicates there is significant publication bias in play. More discussion about how these excluded programs differ (in addition to exclusion based on lack of evaluation) is in order. [page 24; lines 46-52]	Thank you for your comment. We agree that the grey literature is important in capturing the evidence. Our initial search of grey literature indicated that primary data evaluating interventions were either not published in the public domain, or were missing, and therefore did not meet our criteria for inclusion in the review. Furthermore, we felt that a mapping methodology would be more appropriate to summarise this evidence-base. We have revised the limitations section to be clearer that our findings do suggest significant publication bias and why we did not include the grey literature in the review: Discussion [page 24; lines 46-52] “A limitation of this study was the omission of grey literature. Preliminary grey literature searching found over 50 community-centred services, which confirms that most relevant interventions reside outside publication in peer review journals and indicates significant publication bias. We were unable to include these interventions in this review due to the limitation and availability of primary data. Furthermore, we consider a mapping methodology to be more appropriate to comprehensively summarise this evidence(36).”
Reviewer 2: Katarzyna Karolina Machaczek, Sheffield Hallam University, Advanced Wellbeing Research Centre	
Abstract: Some BAME groups’ I suggest the authors provide more details here. Some BAME groups such as... [Page 3]	Thank you for your comment. We have added the ethnic groups that were included in our search strategy but not targeted in any studies. The sentence is now: “Knowledge gaps emerged around effective interventions for men, populations such as Chinese, Arab and Travellers, and tackling the wider determinants of mental health.”
Abstract: Strengths and limitations of the study Line 31 “This scoping review addresses the large evidence gap which exists on the effectiveness of community-centred	Thank you, we have changed the word from whom to who.

interventions in improving public mental health of UK black and minority ethnic (BAME) populations, whom are disproportionately affected by poor mental health'. For 'whom' rather than 'who' to be right in this sentence the authors would need to say "poor mental health affected whom" or something like that. I suggest the authors change 'whom' to 'who' – it's always more idiomatic. [Page 3, line 42]	
Abstract: Please add hyphen [Page 3, line 47]	Thank you, this has been addressed.
Introduction: "Further, the long-standing societal disadvantage and discrimination towards these groups has demonstrable consequences on mental health outcomes, for example through overrepresentation in the criminal justice system (13); lower uptake and poorer satisfaction with care(14); and poorer recovery rates and less likelihood to be in employment(15,16)." This sentence could be improved by dividing the mechanisms that may lead to poor mental health outcomes (e.g. lower uptake of treatment) and outcomes themselves (e.g. poorer recovery rates). [Page 4, lines 3-17]	Thank you for your comment. We have revised the introduction to make clearer the distinction between risk factors, mechanisms leading to poor mental health, and outcomes.
Methods: by involving stakeholder and peer researchers as part of a wider research programme (36). – please change 'stakeholder' to 'stakeholders' [Page 6, line 5]	Thank you. We have made this revision.
Methods: I'm curious whether the authors used appropriate frameworks (e.g. PICOS) to the literature search process? Table 1 suggests they did. Could the authors add a sentence or two to explain this – to aid clarity for the reader? [Page 6, lines 12-16]	Thank you for your comment. We have updated the text to better reflect the PICOC framework that was used to develop our search strategy. The revised sentence reads: "The search strategy (Appendix 1) was created with the support of a medical librarian and was based on the PICOC (population; intervention; comparison; outcomes; context) framework. It included key terms for ethnicity, age range, geography and mental health outcomes, but no intervention and comparison terms were included to optimise capture of all relevant studies."

Methods: Literature searches: Please provide justification for a start date of 1990 [Page 7, lines 26 -28]	We excluded studies before 1990, as we wanted the review to reflect the current context in the UK. Additionally, systematic data collection on ethnic classification in the UK only began with the 1991 census. We have updated the text to: “We included studies published 1990 onwards so that our findings would inform contemporary policy and practice. Furthermore, systematic ethnicity data collection in the UK began with the 1991 Census(37)”
Results: “All studies based on primary research,” please remove comma [Page 8, line 55]	The comma has been removed in Table 1.
Results: Please change “participant’s’ to ‘participants’ [Page 17, line 24]	We have made the necessary change.
Results: please correct punctuation in this sentence: “There was significantly greater improvement in social functioning in the social intervention and combined treatment groups at three months (6.1, 95% CI: 1.4,10.8 and 5.9, 95% CI:1.5,10.2), however this effect did not persist at 9 months.” [Page 18 Lines 43-45]	Thank you. We have edited the sentence to improve the punctuation and clarity. “Social functioning significantly improved in both the social intervention and combined treatment groups at three months (6.1, 95% CI: 1.4,10.8 and 5.9, 95% CI:1.5,10.2). However, this effect did not persist at 9 months.”
Results: “Jacob et al. (56) conducted an RCT in one general practice in Ealing, London evaluating the effect of a culturally-appropriate educational leaflet on depression for women of Indian, Bangladeshi and Pakistani heritage (n=70).” there is a coma missing in this sentence [Page 19, lines 71-73]	Thank you. We have revised the sentence for clarity and have removed the repetitive detail as per Reviewer 5’s comment: “Jacob et al.(56) evaluated the effect of a culturally-appropriate educational leaflet on depression delivered in a general practice to women of Indian, Bangladeshi and Pakistani heritage (n=70).”
Results: “Although the authors suggest a financial barrier es that inhibits people joining the gym, no formal data were collected on cost or participant finances.” ‘es’ seems to be redundant [Page 20, line 13]	“es” has been removed.

Discussion: Please add 'the' before "World Health Organisation" please add "s" to increase ... mental health services increase [Page 24 line 35-38]	Thank you. We have added "the" and "s". We have also revised the following sentence for clarity: "Evidence suggests that mental health services that are provided by the voluntary and community sector and embedded in communities increases trust among BAME communities. This in turn promotes awareness of mental health problems and access to mental health services (16)."
Reviewer 3: Masahito Fushimi, Akita University Health Center, Akita University, Japan	
General Comment: Regarding the issues related to the generalizability of the study results, since BAME covers a broad and diverse range of groups, it is difficult to conduct research in this field under a unified design. Therefore, as the authors correctly state, it is difficult to find high-quality evidence. I think this problem also applies to the results of this study. In other words, I think it is difficult to use the results of this study as evidence applicable to the many diverse minorities worldwide.	Thank you for your comment. We agree and have made the following revisions to address the limited generalisability of this research:  - Introduction (Page 4 lines 18-22) we have added the following sentence to critique the term BAME: "The term has been criticised as grouping together a heterogenous population with diverse health needs (19). Evidence indicates differential and complex risks for poor mental health(20)" - Introduction (Page 4 Line 19): we referenced a recent BMJ editorial titled "The Language of Ethnicity, by Khunti et al." that argues for the collective terms BAME and BME to be abandoned. - Discussion (Page 25 lines 58-60): We have added the following, "A further limitation is the generalisability of our results that include all BAME populations in the UK; homogenous recommendations cannot be made to a culturally and ethnically heterogenous population."
Methods: In this study, a significant amount of literature was extracted in order to provide recommendations to those conducting intervention studies. Meanwhile, many studies were not included in this present study because of problems such as the scale of the study and the observation period. I think that it	Thank you for this comment. The scale and observation periods of the studies were not used as a basis for exclusion. This study was only reviewing peer reviewed literature (the reasons for excluding grey literature are outlined on page 24; lines 46-50). All study designs that fit our eligibility criteria (Table 1, page 8) and involved

will be a valuable contribution to intervention studies in the future if you describe and advise on the size, duration, and type of study designs by which intervention studies may be conducted. I further suggest you present these recommendations to researchers from the standpoint of scoping reviews. I suggest you clarify this point in your paper.	primary data collection were included, regardless of size or duration. I hope that this clarifies your concern.
Discussion: The third part of your Discussion states that lay health workers' activities can help reduce stigma. While I agree with this position, the possibility that non-professionals (including lay health workers) may have more stigma than professionals cannot be ruled out. Based on this viewpoint, it is also necessary to cite any literature that identifies the danger of intervention by non-professionals.	Thank you for your comment. To address your comment, we have added the following sentence [page 24; lines 28-30]: “However, lay health workers, unless they have received anti-stigma training, may hold beliefs that are more stigmatising than those held by health professionals(72)”.
Discussion/ limitations: This study excludes new migrants and refugee groups. Although this decision was informed by the high prevalence of PTSD and other conditions in these groups, it is widely regarded that a high incidence of traumatic experiences is the general case among BAME groups. Moreover, new migrants and refugees also require not only professional cure but care (from lay health workers, etc.); hence, it is appropriate to consider whether their exclusion was necessary or not. [Page 25 lines 66-72]	Thank you for your comment. We have elaborated on our reasons for excluding new migrants and refugees. We hope that this revision is enough to ease your concerns on the exclusion of these groups. We have written, “Interventions specifically targeting refugees, asylum seekers and new migrants were excluded. In the UK, for example, refugee’s makeup approximately 0.2% of the population and new migrants 1.1% and research indicates that these groups are more likely to require specialised clinical interventions focussed on reducing psychological trauma(78,79). However, in light of the overall paucity of evidence in this field and the likelihood of shared need, future research should consider whether this exclusion is necessary.”
General comment: On a minor note, you use the term “black and minority ethnic (BAME)” in the Strengths and Limitations section of your study, while in other parts you use the term “Black, Asian ,and Minority Ethnic (BAME)”. This should be corrected for uniformity and accuracy.	We have made this correction.

Reviewer 4: Haider Mannan, Western Sydney University, Australia	
General comment: This article on scoping review is in an area where there is a gap in the relevant area for UK. The article is stated this gap. The aims are clear also. The structure of the article is ok. The pros and cons of the study are also well addressed. However, I feel the writing could be improved. Therefore, i suggest a resubmission with a minor revision. The article requires proof reading before resubmission.	Many thanks for your comment. We have revised the article throughout to address any grammatical errors, as well as make improvements in its clarity.
Reviewer 5: Genevive Meredith, Cornell University, USA	
Introduction: Consider adding "medical" before care [Page 5, line 23]	Thank you for your comment. We have revised the introduction to better reflect the racial and ethnic challenges faced by BAME groups in the UK. As a result, this sentence is no longer included.
Discussion: consider adding assessment of grey literature to next steps - could be valuable info re: intervention selection, processes used, and facilitators or barriers [Page 27, lines 6-8]	Thank you for this suggestion. We agree with Reviewer 5 that grey literature searching should be undertaken as part of the next steps. We have acknowledged the need for this in our conclusion Page 27 line 6-8: "Our next steps are to map the promising community activities and interventions that are currently being provided to help identify emerging evidence."
Methods: May be worth noting why translation of studies from other countries akin to UK was not considered [Page 7, lines 28-31]	Thank you for this suggestion. We have added the following sentence to our methods: "We included only UK studies due to variation in how race, ethnicity and ancestry are represented in different countries. This is a consequence of each countries' unique pattern of migration and its political and social context."
I found this sentence hard to understand - may be missing a word, or re-work punctuation?	Thank you. We are uncertain as to which sentence this comment relates to. However, we have reviewed the entire paper for its clarity and

	hope that this comment has now been addressed.
Methods: (II) Study selection - should you list how many articles were identified via the web search, even though you also list in results? I ask because you note a % in the next pp. [Page 9, lines 68-69]	Thank you for your suggestion. We have added the number of articles identified via web search: “Two independent reviewers (GZ,CB) screened non-duplicate titles and abstracts against eligibility criteria (n=4501).”
Methods: say more here? screened titles and abstracts for what? what made an article eligible for full review? When did SG come in (title/abstract or full review?) [Page 9, lines 68-71]	Thank you for suggesting more detail here. We have made the following changes to clarify our screening process: “Two independent reviewers (GZ,CB) screened non-duplicate titles and abstracts against eligibility criteria (n=4501). The abstracts that matched criteria (table 1) were reviewed in full by GZ and CB (n=45). All conflicts were resolved by a third reviewer (SG); this was needed for 31% (n=14) of full text articles.”
Methods: presumably duplicates were removed before reading all? [Page 9, line 68]	To clarify that duplicates were removed before screening, we revised the following sentence: "Two independent reviewers (GZ,CB) screened non-duplicate titles and abstracts against eligibility criteria (n=4501)"
Methods: you note getting from 45 studies to 7. How did this happen? Not clear from methods section. [Page 12, lines 3-4]	Thank you for your comment. We have added a sentence to clarify that we went from 45 to 7 studies after full-text review because 38 studies did not match our eligibility criteria: “Seven out of the 45 studies reviewed in full-text matched eligibility criteria and were included in this review”

Results: add a paragraph about the demographics of people in the 7 studies? Later you note more women than men - may be worth stating up front. [Page 12, lines 13-15]	Thank you for your comment and we agree with Reviewer 5 that it makes sense to state that our results had a predominant focus on women. We have added the sentence: “Four studies specifically targeted women and five studies specifically targeted participants with anxiety or depression.”
Results: heterogeneous in what respect? Also, maybe add a sub-heading of 'intervention type and focus' here? [Page 12, line 9]	Thank you for this suggestion, we have added a sub-heading titled "Intervention type and focus" and clarified that the interventions were, "highly heterogenous in design and focus".
Results: given the small number of studies, consider phrasing this in another way. one had.... three had... the majority had... the median or mean was... [Page 13, lines 26-28]	Thank you for this suggestion. We have added detail into the sentence below: "All studies had limitations relating to small sample sizes (n=9-123, mean=43, median=30), short follow-up periods (less than nine months), and participant recruitment.”
Tables 3 & 4 put the article's reference number in column 1 for ease of reference? (page 14 &16)	We have now included reference numbers in Tables 3 and 4.
Methods: you show three categories of outcome measures in this section (addressing.... Promoting... Lay...) - how did you decide on these? Emergent coding? maybe describe in methods? Is this narrative a synthesis of the papers, or an independent (your) assessment of the data and the bias? May want to make that clear. [page 11; lines 93- 95)	Thank you for this insight. The three categories reflect themes derived from the narrative synthesis and discussions with the lead authors. We believe they represent the characteristics that are purported to be useful for improving mental health outcomes in the studies. We have revised the methods to read, Page 11, lines 93-95: "A narrative data synthesis of the papers was conducted by identifying themes and mechanisms common to the community-based interventions.”

Results: (It almost reads like a narrative of the table - not sure that it adds much value?)  might you re-frame this section per your research question: did this intervention improve the mental health and wellbeing of population: y/n, why/why not, lessons to take away [Page 11, lines 127-129]	The bias was determined using validated tools: the Cochrane tool for risk of bias for RCTs, the Cochrane Qualitative and Implementation Methods Group guidance for qualitative papers, and the Mixed-Method Appraisal Tool (MMAT) for mixed methods studies. This is discussed in page 10, lines 90-93. In response to Results: Thank you for this helpful comment. Our research aim was to the summarise the evidence in this field and we believe describing the characteristics of the interventions is an important element. Unfortunately, the level of data analysis that we can provide is constrained due to the quality of the studies included. The format and focus of the studies were also very heterogenous making comparisons difficult. However, we have revised our results to remove descriptive detail that is provided in the table and to reframe in terms of if the intervention improved mental health and wellbeing.
Results: somehow I seem to want to see reference to your study papers also shown here in support of the claims/findings you note. Which studies showed interventions aimed at addressing social isolation? etc. Also, I think this section describes the characteristics of interventions... Was that the focus of your research? Or, were you looking at what types of interventions actually improve MH? If the latter, perhaps organize the discussion to compare and contrast the characteristics between interventions that work vs. don't?	In response to references: We agree and have included references in the results when we refer to findings. In response to results: We hope that we have successfully addressed your concern in comment 34 above.
Additional journal comment	
Authors must include a statement in the methods section of the manuscript under the sub-heading 'Patient and Public Involvement'. This should provide a brief response to the following questions:	This has been addressed in the methods section, and includes the following statement under 'Patient and public involvement':

 • How was the development of the research question and outcome measures informed by patients' priorities, experience, and preferences? • How did you involve patients in the design of this study? • Were patients involved in the recruitment to and conduct of the study? • How will the results be disseminated to study participants? • For randomised controlled trials, was the burden of the intervention assessed by patients themselves? • Patient advisers should also be thanked in the contributorship statement/acknowledgements. If there is no patient involved in the study, please state "No patient involved" under the sub-heading 'Patient and public involvement'.	"The research question was informed by peoples' experiences and stakeholder workshops. This study did not involve the recruitment of patients, and no patients were involved in the design, or conduct of the study."
---	--

VERSION 2 – REVIEW

REVIEWER	Nhi-Ha Trinh Massachusetts General Hospital
REVIEW RETURNED	27-Jan-2021

GENERAL COMMENTS	Thank you very much for your consideration of our comments.
---

REVIEWER	Dr Katarzyna Karolina Machaczek Sheffield Hallam University, UK
REVIEW RETURNED	04-Feb-2021

GENERAL COMMENTS	The revised manuscript is much improved with the authors making every effort to address the reviewers comments. It is a much stronger and more robust piece as a result.
--

REVIEWER	Masahito Fushimi Akita University Health Center, Akita University, Japan
REVIEW RETURNED	29-Jan-2021

GENERAL COMMENTS	The author carefully examined the matters I pointed out last time (previous reviewer comments), and I confirmed that the necessary corrections were made. I thought this article is a good treatise for publication. Thank you for responding appropriately to my requests. The study is based on the results extracted from a vast amount of literature concerning minorities. This will make several useful contributions to the study of minorities.
---

REVIEWER	Genevive Meredith Cornell, USA
REVIEW RETURNED	06-Feb-2021

GENERAL COMMENTS	Page 2; line 57. "Studies included mostly Indian and Pakistani ethnic groups." seems odd. Maybe 'study populations were ethnically heterogeneous?' ** See below, too. If you add this to discussion/recommendations, perhaps you can discuss why this might be the case? Page 4: Lines 18-23. Paragraph seems out of place. Better suited in 'recommendations'? (studies that do a better job of disaggregating race/ethnicity?) Page 8: Table 1: Why were adults with no known mental disorders included? Page 11: Lines 13-14: What do you mean by "there was a specific focus of interventions for women (n=4) and people with anxiety or depression (n=5)? Describe better, note that these are not mutually exclusive, etc. Page 11: Line 17: add (n)s for number of studied with single session vs. other lengths Page 11: Line 18: How many is 'most'? Page 11: Line 22: Seems like this should be a new paragraph? Page 13: Table 3 - put study citation (#) in Author/Year column Page 17: Line 11; Page 18: Line 53; Page 21: Line 110 - I am having trouble tracking what these title are, and how these link to your study's focus: Community-centred interventions for improving public mental health among adults from minority ethnic populations in the United Kingdom: a scoping review. How did you come up with these grouping structures? Were these study questions you hoped to specifically answer (if so, I think this should be stated in the intro/methods?)? Perhaps you can provide a bit more of an intro to this section to anchor the reader? It looks like you are just summarizing each of the 7 studies here, grouped into three sections. I might like to see something more creative here! A way to cross-walk these studies? Or, were they that discrete? Page 24: Line 32+ - compared to points 1-3, this stands out as not specifically referencing any of the reviewed studies. Page 24: Line 40+: This might be a place to place the "Indian and Pakistani ethnic groups" comments, and speculate as to why? Is there more mental illness in those populations? Do they may up more of the U.K. population? Is there a greater sense to community/care? etc. Why are there fewer interventions or studies with other groups? Page 24" Line 50: "We were unable to include these interventions in this review due to the limitation and availability of primary data" - not sure this argument holds?? Did you actually analyze primary data? Your methods don't suggest that. I suggest re-working this paragraph and clarifying your reasoning (simply too many, as you suggest in the intro?)
--

VERSION 2 – AUTHOR RESPONSE

Comment Number	Outstanding BMJ Open Reviewer's comments	Responses
	Reviewer 5: Genevive Meredith, Cornell University, USA	Thank you for all your comments in strengthening our paper.

1	Abstract: "Studies included mostly Indian and Pakistani ethnic groups." seems odd. Maybe 'study populations were ethnically heterogeneous?' ** See below, too. If you add this to discussion/recommendations, perhaps you can discuss why this might be the case? [Page 2; line 57]	Thank you for your comment. We have revised the sentence to read "study populations were ethnically heterogeneous and targeted people mainly from South Asia." We have responded why 5/7 studies targeted ethnic groups from South Asia in the later comment raised.
2	Introduction: Paragraph seems out of place. Better suited in 'recommendations'? (studies that do a better job of disaggregating race/ethnicity? [Page 4: Lines 18-2]	Thank you for your comment. Our reasoning for introducing a paragraph on ethnicity terms in the introduction was our hope that the reader would be aware that the umbrella term of BAME gives an impression that this is a homogenous group, when in fact there are differential mental health risks between groups. However, we have reviewed and revised the paragraph.
3	Methods: Why were adults with no known mental disorders included? [Page 8: Table 1]	The scope of our review was interventions aimed at public mental health, that is, the promotion of mental health and wellbeing and the prevention of mental illness. Thus, adults with no known mental disorders were included within this population definition.
4	Results: What do you mean by "there was a specific focus of interventions for women (n=4) and people with anxiety or depression (n=5)? Describe better, note that these are not mutually exclusive, etc. [Page 11: Lines 13-14]	Thank you, we have rephrased the sentence to read, "Certain subgroups were targeted more than others with interventions specifically tailored to individuals from Indian or Pakistani heritage(n=3), women(n=4), and people with anxiety or depression(n=5)." We hope that this clarifies your point that these are not mutually exclusive groups.
5	Results: add (n)s for number of studied with single session vs. other lengths. [Page 11: Line 17]	Thank you for your comment. We have revised the sentence accordingly.
6	Results: How many is 'most'?[Page 11: Line 18]	This sentence has been revised, "Four studies aimed to improve mental health by expanding social networks and facilitating social support (51,53,54,57)"
7	Results: Seems like this should be a new paragraph? [Page 11: Line 22]	We are happy to accept this suggestion.
8	Results: put study citation (#) in Author/Year column [Page 13: Table 3]	We have added study citations in the Author/year column

9	Results: I am having trouble tracking what these title are, and how these link to your study's focus: Community-centred interventions for improving public mental health among adults from minority ethnic populations in the United Kingdom: a scoping review. How did you come up with these grouping structures? Were these study questions you hoped to specifically answer (if so, I think this should be stated in the intro/methods?)? Perhaps you can provide a bit more of an intro to this section to anchor the reader? It looks like you are just summarizing each of the 7 studies here, grouped into three sections. I might like to see something more creative here! A way to cross-walk these studies? Or, were they that discrete? [Page 17: Line 11; Page 18: Line 53; Page 21: Line 110]	Thank you for your comment. We identified three categories and themes (addressing social isolation and loneliness; promoting access and use of services; delivered by lay health workers) through a narrative synthesis of the 7 studies and research team discussions. We framed these themes through a practice, policy and research lens as potentially useful building blocks for future public mental health interventions for BAME groups. Unfortunately, we were limited in making comparisons due to the high heterogeneity and poor quality of the studies and did not want to overstate our interpretation. To help anchor the reader, we have added a sentence before the section that reads, "Following narrative synthesis, we identified three themes with the perspective of informing the delivery of public mental health interventions for BAME communities: addressing social isolation and loneliness, promoting access and use of services, and being delivered by lay health workers." [page 16; lines 11-12]
10	Discussion: compared to points 1-3, this stands out as not specifically referencing any of the reviewed studies. [Page 24: Line 32+]	Thank you for your comment we have referenced these studies.
11	Discussion: This might be a place to place the "Indian and Pakistani ethnic groups" comments, and speculate as to why? Is there more mental illness in those populations? Do they may up more of the U.K. population? Is there a greater sense to community/care? etc. Why are there fewer interventions or studies with other groups? [Page 24: Line 40+]	Thank you for this comment. We are unsure as to why there is a focus on South Asian communities. It may reflect that they make up a greater proportion of the UK population proportionally when compared to other ethnic groups; people of Indian, Pakistani and Bengali heritage makeup roughly 5% vs 3.3% for Black ethnic groups. However, we chose not to speculate on this point in the discussion as we considered that a key message was the total number of studies were few for all ethnic groups. This could point to structural racism but felt that this requires further unpacking and research. We were also constrained by word count.

		However, we have added the sentence, “Of the community interventions we identified, we found more targeted South Asian ethnic groups, which may reflect these groups making up a greater proportion of the UK population.”
12	Discussion: "We were unable to include these interventions in this review due to the limitation and availability of primary data" - not sure this argument holds?? Did you actually analyze primary data? Your methods don't suggest that. I suggest re-working this paragraph and clarifying your reasoning (simply too many, as you suggest in the intro?) [Page 24" Line 50]	Thank you for your comment. We agree that it should be clearer that it was both the number of interventions and the availability of primary data that meant we were unable to include these interventions. We have made the following revisions: [page 6; lines 17-23] “We excluded evidence from the grey literature as an initial search of primary data from local government, relevant third sector and NHS websites identified numerous activities and possible interventions (over 50 in a small geographical area). Information was presented in formats such as a flyer or a website or Facebook page describing services and activities but with limited descriptions of the community intervention and outcomes data. Consequently, data synthesis exceeded the methodological approach of a scoping review; a mapping methodology would be more appropriate (36).” [page 23; lines 49-50] “We were unable to include these interventions in this review due to a large number of individual activities/interventions and the lack of available primary data. We consider a mapping methodology to be more appropriate to comprehensively summarise this evidence”